# On the Detection and Functional Significance of the Protein–Protein Interactions of Mitochondrial Transport Proteins

**DOI:** 10.3390/biom10081107

**Published:** 2020-07-25

**Authors:** Youjun Zhang, Alisdair R. Fernie

**Affiliations:** 1Center of Plant Systems Biology and Biotechnology, 4000 Plovdiv, Bulgaria; 2Max-Planck-Institut für Molekulare Pflanzenphysiologie, Am Mühlenberg 1, 14476 Potsdam-Golm, Germany

**Keywords:** protein–protein interaction, inner mitochondrial membrane, mitochondrial carrier family

## Abstract

Protein–protein assemblies are highly prevalent in all living cells. Considerable evidence has recently accumulated suggesting that particularly transient association/dissociation of proteins represent an important means of regulation of metabolism. This is true not only in the cytosol and organelle matrices, but also at membrane surfaces where, for example, receptor complexes, as well as those of key metabolic pathways, are common. Transporters also frequently come up in lists of interacting proteins, for example, binding proteins that catalyze the production of their substrates or that act as relays within signal transduction cascades. In this review, we provide an update of technologies that are used in the study of such interactions with mitochondrial transport proteins, highlighting the difficulties that arise in their use for membrane proteins and discussing our current understanding of the biological function of such interactions.

## 1. Introduction

The mitochondrion is an important cellular organelle involved in cellular respiration via the tricarboxylic acid (TCA) cycle and oxidative phosphorylation (OXPHOS), fatty acids biosynthesis [1,2,3,4], photorespiration [5], compartmentation of metabolic processes such as amino acids biosynthesis [6,7], and C4 photosynthesis [8]. To perform roles in these many processes, eukaryotic mitochondria exchange metabolites with the cytosol and other organelles through both the outer mitochondrial membrane (OMM) and inner mitochondrial membrane (IMM). The outer membrane has long been believed to allow the diffusion of ions and uncharged small molecules though porous membrane proteins, especially the voltage-dependent anion channel (VDAC) [6,9]. 

Interestingly, as a convergence point, VDACs interact with diverse partners, including those that are broadly functionally classified as apoptosis-, cell signaling-, and cytoskeleton-related proteins, as well as metabolic enzymes [10]. For larger molecules, including proteins, to be translocated involves three machineries: the sorting and assembly machinery (SAM complex), the translocase of the outer membrane (TOM complex), and the mitochondrial distribution and morphology (MDM) complex [11]. Moreover, recent research has suggested that several outer membrane channels and their transport mechanisms are more diverse than originally thought [12], such as the acyl-dihydroxyacetone phosphate reductase (Ayr1) and two additional anion-selective channels (OMC7 and OMC8), which have been suggested to transport metabolites and ions. However, in all cases, detailed information concerning transported substrates is lacking [12,13]. In addition, several cytoskeleton proteins, such as tubulin, vimentin, plectin, and desmin, have been reported to interact with the mitochondrial outer membrane, where they are suggested to be involved in the ATP/ADP transmission control through VDAC, thereby mediating or influencing mitochondrial function [14]. 

In sharp contrast, molecules and ions can only pass across the inner mitochondria membrane by the aid of particular membrane transport proteins, each of which is selective for a specific molecule or ion. Around 180 mV of an electrochemical gradient is generated across the inner mitochondrial membrane during the operation of oxidative phosphorylation. Protein translocases of the outer (TOM) and inner (TIM) membrane form a supercomplex spanning the intermembrane space and appear to be held together by the polypeptide in transit [15]. In addition, a wide array of metabolites cross the inner mitochondrial membrane by the nuclear-encoded molecular gatekeepers of the mitochondrial carrier (MC) family [16]. 

Featuring six conserved transmembrane α-helical regions [17], most mitochondrial carrier family proteins (MCF) are relatively small, around 300 amino acids in length, ranging from 30 to 35 kDa [18]. The primary structure of most MCs is comprised of three homologous regions, each containing around 100 amino acids [19], and with both the N and C terminals exposed to the intermembrane space [20,21]. Every repeat region is suggested to include two transmembrane segments flanking a short helical region that is parallel to the lipid bilayer [22]. 

In addition, every repeat region is proposed to have two hydrophobic transmembrane segments connected by a long hydrophilic matrix loop [23,24] and the structural motif PX[D/E]XX[K/R]X[K/R] (20–30 residues), [D/E]GXXXX[W/Y/F][K/R]G (IPR00193) [16]. Of the MCFs, the ADP/ATP carriers, which import ADP from the cytosol and export ATP from the mitochondria, are the best investigated and our current understanding of MCFs is largely derived from the crystal structures of different versions and conformations of this carrier [25]. The alternating access mechanism (previously named ping-pong mechanism) of the transport cycle includes substrate binding to the carrier in its c-state (matrix-side closed and cytoplasmic-side open state), undergoing a conformational change to a transition state, and finally relaxation to the m-state (cytoplasmic-side closed and matrix-side open state), followed by the release of the substrate into the mitochondria. In the m-state, the counter-substrate binds to the carrier, undergoes a conformational change and eventually converts to the c-state, and releases the counter-substrate into the cytosol. After release of the counter-substrate, the carrier can start the transport cycle anew. Opened and closed carriers expose the substrate-binding site to one or the other side of the membrane with alternating access ping-pong mechanisms [25,26,27,28,29,30,31,32,33]. This proposed mechanism was initially based on studies of the high-affinity inhibitor ligands of the ADP/ATP carrier, carboxyatractyloside and bongkrekic acid, and their competition with the transported substrates ADP and ATP [29,30]. 

MCFs mediate the transport of carboxylic acids, fatty acids, amino acids, inorganic ions, cofactors, and nucleotides across the mitochondrial inner membrane across the kingdoms of life [26], and are crucial for many cellular processes [17,34,35,36]. Given that typical detergent application reveals that MCFs are small in size, the formation of the homomeric and heteromeric complexes may simply help to assemble the transporting complex and avoid random, unfavorable protein–protein interactions in the crowded environment of the inner mitochondrial membrane. These reports, including the tricarboxylate carrier, the dicarboxylate carrier, and the oxoglutarate carrier (Table 1), are, however, as we describe below, erroneous, further highlighting the need for carefully controlled experiments using multiple methods. However, the structural folds of the ADP/ATP carriers were proved to function as monomers [37,38]. MCs are now thought to exist and function as monomers. The only carrier definitely known until now to exist as a homodimer is the human aspartate-glutamate carrier (with 2 isoforms: AGC1 and AGC2). This carrier consists of two domains: an N-terminal soluble regulatory domain (with four calcium-binding sites) and a C-terminal MC catalytic domain [39,40]. Evidence has been provided that two molecules of the aspartate-glutamate carrier are linked together by their N-terminal regulatory domains [41].

The association of mitochondrial transport proteins plays an important role in the movement of metabolites across the IMM. For example, although not a member of the MCF, the heteromeric complex of the mitochondrial pyruvate carrier (MPC) is strategically positioned at the intersection between glycolysis in the cytosol and OXPHOS in the mitochondria [48]. Similarly, the association of the ATP synthasome, F0F1-ATPase, complex of adenine nucleotide translocator (ANT), and the phosphate carrier (PiC), facilitates a mechanism for adenine nucleotide and pyrophosphate release [56,57,58,59]. In addition, the interaction between the MCF and mitochondrial proteins may also improve metabolite import; for example, the dicarboxylate carrier (DIC) interacts with malate dehydrogenase, acting as an oxaloacetate shuttle and thereby improving functional coupling of the citric acid cycle with the shuttle [60]. Despite their importance in assisting metabolite transport, MCF protein interactions have been poorly studied in living cells—most likely due to methodological limitations. Indeed, although various protein interaction approaches to investigate membrane–protein interactions have been developed over several decades, such as coimmunoprecipitation and chemical cross-linking, bioluminescence resonance energy transfer (BRET), blue native polyacrylaminde gel electrophoresis (BN-PAGE), yeast two-hybrid (Y2H), and fluorescence resonance energy transfer (FRET), only a few have been employed for studying the interactions of mitochondria carriers [48,61,62,63]. Here, we review such protein–protein interactions of the mitochondrial carrier family and discuss the major challenge of acquiring information about the interactions of integral membrane proteins. Despite the relative paucity of information on such interactions, we argue that they are likely underappreciated across the kingdoms of life and may represent an important constituent of compartmental and regulation of metabolism. In the following review, we detail historical and contemporary observations of protein interactions of MCF, detailing methods by which they are determined, and suggesting possible biological implications of such assemblies.

## 2. Experimental Evaluation of the Protein–Protein Interactions of Mitochondria Carriers

Protein interactions of the membrane play an important role in several biological systems, such as regulation of metabolic pathways, signal transduction cascades, regulating metabolite import, and the formation of membrane complexes [64]. The cumulative databases of membrane protein complex structure provide considerable insight into how folding and packing of their transmembrane segments contribute to their diverse functions [65,66]. Single-span transmembrane helices have been suggested to act as anchors for more “interesting” water-soluble domains or merely to constitute models for multipass protein folding, such as leucine zippers, that mediate protein complex association in water-soluble proteins. Transmembrane proteins can additionally direct protein–protein interactions within the membrane and participate in signal transduction across lipid bilayers [64]. Thus, it is of particular importance to prepare the protein components such that the structure, topologies, and functions remain intact prior to their analysis. Given that various diverse types of living cells, such as yeast or tissue culture, are widely used to investigate in vivo mitochondria carrier interaction, it is important to choose the correct “horse for the course” in order to maximize the insight that can be achieved. In the following section, we give a brief overview of the available tools.

A major breakthrough methodology was the use of bioluminescence resonance energy transfer (BRET), which can investigate protein–protein interactions in live cells based on the enzymatic activity of luciferase (Figure 1b) [48,67]. For example, regarding signal transduction cascades associated with the outer membrane, interactions between retinoic acid-inducible gene I-like receptors and mitochondrial antiviral signaling (MAVS) are known to trigger the immune signaling pathway [68]. Thus, the interaction between RLR and MAVS results in the multistep structural changes from inactive to active states. However, these signal transduction processes are transient states, and as such, extremely difficult to reconstitute in other membrane systems or in vitro experiments [67,68]. For this reason, several studies have rather used the BRET system to provide insight into the structural transition of MAVS between active and inactive conformation in vivo [67,68,69]. In this system, either the N-terminal-fused Rluc- or YFP tags of MAVS could beautifully monitor the interaction between MAVS molecules via a BRET saturation assay. This system is thus strongly recommended to detect the interaction between cytosol proteins and OMM proteins. A further example of the power of this technique came from combining biomolecular fluorescence complementation (BiFC) with the BRET system, which was used to demonstrate that activated MAVS is a highly ordered oligomeric complex and that three proteins associated on the mitochondrial surface [68].

Yet another example is the protein–protein interaction of the mitochondrial pyruvate carrier, which represents a central node of carbon metabolism [48]. The mammal heteromeric complex is composed of two paralogous subunits, MPC1 and MPC2, which regulate pyruvate uptake. In order to monitor the activity and complex formation of the MPC in real time, BRET experiments combining MPC2-RLuc8 and MPC1-Venus were used to detect the assembly of the MPC complex in the presence of various metabolites, including pyruvate lactate, malate, and citrate (Figure 1b) [70]. Using a modified BRET protocol, the authors could monitor the lower MPC activity in cancer cells, which are believed to mainly rely on glycolysis to produce ATP, a characteristic known as the Warburg effect [48]. Intriguingly, this effect was recently shown to be at least partially mediated by the tripartite motif containing-21-dependent ubiquitination and subsequent degradation of phosphofructokinase based regulation of pyruvate kinase turnover in response to cytoskeletal monitoring of cellular tension [71,72]. Returning to the study above, this low activity could intriguingly be turned over by increasing the concentration of cytosolic pyruvate, thus enhancing oxidative phosphorylation. This demonstration shows that the biosensor represents a unique tool for investigating carbon metabolism and bioenergetics under a diverse range of conditions [48,67,68,69]. Importantly, we present several examples in which the BRET system was used for monitoring the interactions of mitochondrial carriers and which should be adopted in future studies. Moreover, combining BRET with BiFC can monitor three way protein–protein interactions at the cell membrane, whilst combining the BRET and FRET—called SRET—represents an attractive strategy by which to understand heteromerization complex in a physiological environment [73]. The application of such methods to other organelle-bounded molecules are detailed below.

Another widely used method is blue native polyacrylamide gel electrophoresis (BN-PAGE) (Figure 2), which can be employed for one-step purification of protein complexes from total cell and tissue homogenates or from biological membranes, including the mitochondrial membranes. It can also be employed to determine physiological protein–protein interactions and to discover native protein masses and oligomeric states. Native protein complexes are recovered from gels by electroelution or diffusion and are investigated by native electroblotting and immunodetection or by in-gel activity assays or employed for 2D crystallization and electron microscopy. Several studies using this method have reported that mitochondrial carriers are dimeric (composed of two ~32 kDa monomers) and, in some cases, can form physiologically relevant associations with other proteins [74,75,76]. For example, the conserved mitochondrial inner membrane proteins migrate on BN-PAGE as a large complex of ~150 kDa [75] or the ~30 kDa MPC dimers, due to different amounts of lipids and detergent bound to MPC [77]. Interestingly, two different heterodimers contain either Mpc1 and Mpc2 (MPC_FERM_) or Mpc1 and Mpc3 (MPC_OX_) were detected by BN-PAGE and were suggested to depend on the carbon source in yeast cells [70]. MPC_OX_ has higher pyruvate transport activity than MPC_FERM_ owing to differences in the C-terminal region of Mpc2/Mpc3. Moreover, the yeast mitochondrial ADP/ATP carrier (AAC3) was also reported to vary in a detergent- and lipid-dependent manner (from ~60 to ~130 kDa) that is not related to changes in the oligomeric state of the protein, while these also vary due to binding to different amounts of lipids and detergent. However, several studies have consistently shown that yeast ADP/ATP carriers are monomeric in structure and function [44]. As the mitochondrial respiratory chain supercomplexes have been investigated by BN-PAGE under different conditions [78], this method could be used to detect the dynamics of the mitochondrial carrier protein complex. In addition, several mitochondrial carriers were also investigated by BN-PAGE, such as the 2-oxoglutarate carrier [52], the dicarboxylate carrier [51], and the tricarboxylate carrier [50]. We will discuss these below. 

Size exclusion chromatography or gel filtration is yet another established technique for the determination of protein interactions relying on the ability, or lack thereof, of particles to pass through a column of porous beads according to their hydrodynamic radius (Figure 2). Given the large amount of bound detergent and lipid, accurate values of the molecular mass and dimensions of the protein need to be carefully accounted for in the case of mitochondrial carriers in solution [37]. For example, the Mpc1/Mpc3 complex eluted as a single peak by size exclusion chromatography, indicating that the complex was monodisperse [49]. Using size exclusion chromatography with mass spectrometry, the conserved protein complexes of 13 plant species were identified, including the mitochondrial membrane protein [79,80]. An important advantage of this method is that it is compatible with physiological conditions and thereby allows the investigation of the dynamics of protein complex formation and disassembly. 

Differential tagging and affinity purification have also been used to investigate the association state of mitochondrial carriers (Figure 2). This method is based on the principle that an associated stable complex in detergent should be copurified by affinity chromatography if one of the components of the complex contains an affinity tag. This approach is higher throughput, faster, and more straightforward than other methods, as the amounts of bound lipid and detergents do not need to be measured. However, the denatured mitochondrial membrane may result in false negative and false positive identifications. To avoid nonspecific aggregation, which can be misinterpreted as native interactions, the misfolding or unfolding of the carriers should be prevented by additional steps. Additional detergents, applied at these steps, also affect the intensity of the LC-MS measurement and necessitate further steps to reduce them. This method was used to obtain a preparation enriched in calcium transporters from Triton X-100 extracts of rat liver mitochondria inner membranes [81]. Similarly, when the yeast phosphate carrier Mir1p was expressed in inclusion bodies as a misfolded form in *E. coli* with a FLAG tag [82], the homo- and heterodimeric forms of phosphate carriers, which were combined by two differently tagged unfunctional monomers by affinity chromatography, were full active after reconstitution suggested the phosphate carrier works as dimer [82]. In addition, affinity purification of the plant TCA cycle enzymes also revealed interaction between the IMM proteins citrate synthase, isocitrate dehydrogenase, oxoglutarate dehydrogenase, and the mitochondrial phosphate carrier proteins [83,84,85,86]. Given that affinity purification with mass spectrometry is based on the association of stable complexes, the combination of AP-MS with cross-linking has been suggested to greatly improve detection of transient and unstable protein–protein interactions of the organelle-bound membrane proteins [87,88].

Chemical cross-linking is a classical method used to freeze protein complexes in their native form, and has proven especially useful for capturing transient protein–protein interactions. Given that such reagents can permeate into the cellular compartment in the case of the membrane-permeable cross-linkers, this can be subsequently combined with mass spectrometry (cross-linking coupled with MS: XL-MS) to give vital insight into the spatial arrangement of protein complexes in organelles at the subunit level [89]. However, the complex fragmentation pattern of cross-linked peptides frequently precludes unambiguous identification of the cross-linked peptides. XL-MS is particularly challenging with respect to data analysis. MS-cleavable cross-linkers, for example, disuccinimidyl dibutyric urea (DSBU; containing cleavable C−N bonds) or disuccinimidyl sulfoxide (DSSO; containing cleavable C−S bonds), are recent innovations. The merits of such MS-cleavable cross-linkers are that their cleavage (in the gas phase during MS) produces distinguishable ion doublets, allowing the cleaved peptides to readily be discovered via a database search [89]. The dynamic interactions of coexisting respiratory supercomplexes have been successfully captured using this technique [90]. It was also used to construct the mitochondrial protein interactome of a large-scale model of stable and transient interactions, and interactions between membrane proteins [91]. That said, one limitation of the XL-MS approach is that cross-links within the target cannot be distinguished as intermolecular or intramolecular, for example, as in mitochondrial prohibitin complexes [92].

The XL-MS method could, however, also provide insight into many protein complexes for the reconstruction of low-resolution three-dimensional structures, but it must be considered that the cross-linking process may result in artefactual protein–protein interactions. In the presence of bongkrekic acid (BKA), the bovine ADP/ATP carrier could be fixed in the matrix state in submitochondrial particles (inside-out membranes) by copper-o-phenanthroline cross-linking, while the presence of CATR locked the carrier in cytoplasmic state and prevented cross-linking. In addition, sodium dodecyl-sulphate (SDS) was reported to inhibit either AAC1 or UCP1 interactions via cross-linking [93,94]. This observation has also been used to support the dimer model with the rationale being that the carriers are unfolded in SDS or solubilized to monomers, and thereby are no longer associated as dimers and cannot be cross-linked to their usual interactors. Both the mitochondrial oxoglutarate and phosphate carriers could be cross-linked in detergent, but not in situ in the membrane [37,53]. Taken together, this suggests that cross-linking occurs in different parts of the mitochondrial carriers, suggesting the presence of many interfaces distinct from dimerization. The combination BN-PAGE and size exclusion chromatography with cross-linking could greatly aid in avoiding protein complex disassembly when the complexes are separated from the mitochondrial membrane by denaturing detergents.

Recently developed proximity-dependent labelling methods have been used to detect the transient protein–protein interactions under native conditions in living cells [95]. The basic principle of the proximity-based biotin labelling method or proximity-dependent biotin identification (BioID) is that a protein of interest (bait) tagged by a promiscuous biotin ligase (BirA R118G mutant, from *E. coli*) could convert the biotinylation of nearby endogenous proteins with lysine side chains (within ∼10 nm) after the addition of a biotin supplement to the tissue culture medium as described by Roux et al. (Figure 2) [96]. Due to the covalent biotinylation of the targets, these biotin-labelled proteins are stable following stringent cell lysis treatment with the protein extraction, followed by affinity purification (such as streptavidin beads), and several washing steps. This approach can thereafter be combined with MS measurement to detect biotin-labelled proteins and to screen the protein–protein interaction networks with high spatial resolution in living cells. As such, this approach has been successfully used to evaluate physiologically relevant networks of protein–protein interactions and is especially potent for detecting low-affinity and/or transient associations, such as the interactomes involved in signal transduction [95]. Due to the potential masking of, for example, the mitochondrial targeting sequence, it is, however, important to consider the effects of the BirA fusion tag at either the N- or C-terminus on the biological activity and/or subcellular localization of the target protein. 

In addition, this method was enhanced by targeting of the nearby proteins using a smaller type of biotin ligase (from *Aquifex aeolicus*; BioID2) to link the bait protein, which has the added benefit of decreasing the amount of biotin supplementation that is required. This improved assay has been used to elucidate a mitochondrial macromolecular complex in mammalian cells by fusing the BirA from *A. aeolicus* to the C-terminal of an inner mitochondrial membrane-bound protein prohibitin 2 (PHB2) [97,98]. In this research, the recombinant PHB2 behaved in an almost identical manner to the interacting protein and biotinylated a large number mitochondrial proteins, including its PHB1 isoform, SAM and mitochondrial contact site and cristae organizing system (MICOS) subcomplex [97,98]. Although BioID approaches have been successfully applied to detect the interaction between membrane proteins that exhibit a wide range of subcellular localizations, such as the mitochondria, cell–cell junctions, nuclear lamina, and endoplasmic reticulum, it is still not used for monitoring the interaction of the mitochondrial carriers. Indeed, the BioID approach has an inherent limitation for identifying macromolecular complex in mammalian cells by fusing to prohibitin 2 (PHB2), an IMM-bound protein [97]. Given that both the N and C termini are proposed to be exposed to the intermembrane space [19], the catalytic activity from BirA may not reach the target site of the mitochondrial proteins. Furthermore, if the structural surface of the target protein proximate to the bait lacks lysine residues, this method cannot be used [97]. The combination of proximity-dependent labelling and chemical cross-linking offers great promise to enhance our understanding of multiprotein complexes that are difficult to prepare, such as organelle-bound membrane proteins.

## 3. Proteins Associated to Mitochondrial Outer Membrane Carrier

Although most of the mitochondrial carriers are enriched in the inner boundary membrane of the mitochondria, recent studies suggest that several outer membrane channels can also transport metabolites and ions without further research of their transported substrate [12,13]. The exchange of hydrophilic compounds, such as metabolites and ions across the OMM of eukaryotes is facilitated by specific voltage-dependent anion channels [6,9] and followed by active transport across the IMM mediated by specific nuclear-encoded membrane transporters. The appearance of Donnan potential, which relates to the distribution of ion species between two ionic solutions across the outer membrane, could render the VDAC formation in either the open or closed state. In the open state, VDAC shows a considerable preference for metabolic anions with weak selectivity. In the closed state, VDACs still conduct small cations, such as K^+^, Na^+^, and Ca^2+^, and Cl^−^ by a cation selective pore of 1.8 nm in diameter, while the transition of major anionic metabolites such as ATP, ADP, and respiratory substrates is prevented [99]. Several publications have been reported that the regulation of VDAC conductance contributes to energy metabolism and mitochondrial dysfunction [99,100].

Moreover VDACs have been reported to interact with diverse partners, including proteins of apoptosis, cell signaling, cytoskeleton-related proteins, and metabolic enzymes (Table 1) [10]. For example, glucose-6-phosphate, which is the product of the hexokinase-catalyzed reaction, was suggested to potentiate ATP release from mitochondria with the recovery of normal metabolism and substrate replacement, was even more increased when hexokinase and glucokinase were dissociated following being bound to VDAC [10,42]. Therefore, the effect of hexokinase dissociation from VDAC is the subject of many studies dealing with the development of potent cancer chemotherapies. Similarly, creatine kinase, which also acts as an energy sensor and mediates antiapoptotic effects via VDAC–ANT complexes, exhibits preferential use of mitochondrial ATP. Moreover, interaction of desmin with various contact sites (VDAC, adenine nucleotide translocator (ANT) and mitochondrial contact site complex) affects mitochondrial permeability transition pore (mtPTP) behavior and respiratory function. VDAC forms a barrel comprised of a transmembrane alpha helix and 13-transmembrane beta strands. The resultant beta barrel encloses a large channel (~3 nm in diameter), which in its open configuration is permeable to molecules up to ~5 kDa in size. In vitro studies have revealed that all eukaryotic VDAC channels adopt multiple conductance states. In the IMS, MtCK binds with high affinity to cardiolipin and other anionic phospholipids cross-linking the two peripheral mitochondrial membranes and the ANT, thus forming a complex of MtCK–VDAC–ANT and cardiolipin [43]. The MtCK–VDAC association is enhanced at physiological calcium concentrations. MtCK associates only with the inner membrane and ANT in the cristae space. MtCK preferentially uses mitochondrial ATP that is exported via ANT to phosphorylate creatine, which has a higher diffusion rate in comparison to ATP, thus providing a spatial energy shuttle. The coupling of these partner proteins thereby links VDAC to other cellular functions such as protein import, apoptosis, control of cellular energy metabolism, and tumorigenesis. 

Two clearly distinguishable anion-selective channel activities, OMC7 and OMC8, with pore diameters of roughly 15 Å were detected in vesicles of the OMM. However, as yet, the corresponding channel-forming proteins remain to be identified. In addition, the acyl-dihydroxyacetone phosphate reductase (Ayr1) forms a cation-selective channel. This protein contains a predicted α-helical transmembrane domain and is located in the mitochondrial outer membrane, ER, and lipid particles. However, it is currently unclear whether Ayr1 forms a channel in all these cellular compartments. Several functions in lipid metabolism have been putatively assigned to this protein. Indeed, it is thought to be involved in the biosynthesis of phosphatidic acid, in the mobilization of triacylglycerol, and in the elongation of fatty acids. It contains a predicted Rossman fold and conserved signature motifs such as a NADPH-binding site that are characteristic for the large protein family of short-chain dehydrogenases [13]. Addition of NADPH induces very fast voltage dependent gating of the channel. This voltage-dependent gating occurs unidirectionally. Moreover, in the presence of NADPH, the Ary1 channel, even at high voltages Vm > 100 mV, remained in the open state. However, whether or not the Ayr1 channel is actively and directly involved in lipid biosynthesis remains to be investigated.

In summary, the old concept that the outer membrane functions as a “molecular size-exclusion filter” for the passage of small hydrophilic molecules through VDAC has been strongly questioned by different observations. First, transport processes can occur in cells deficient of all VDAC isoforms [10]. Secondly, the VDAC channel properties are modulated by a variety of different effectors, suggesting that their specificity is under active control [10]. Thirdly, the outer membrane contains, beside VDAC, additional pore-forming proteins that might mediate metabolite flux [10]. Fourthly, the presence of the NADPH-regulated Ayr1 channel indicates that such transport processes across the outer membrane of mitochondria may be subject to redox regulation. In conclusion, metabolite transport across the OMM is more specific and regulated than originally thought, challenging the view that VDAC allows the unregulated passage of all manner of small molecules.

## 4. Proteins Associated to the Inner Membrane of Mitochondria

Metabolites, such as amino acids, carboxylic acids, fatty acids, cofactors, inorganic ions, and nucleotides, are specifically transported across the IMM by members of the mitochondrial carrier family (and other mitochondrially localized transporters that do not belong to the family [101]), to join in many cellular processes. Most metabolites transport as strict counter-exchangers of chemically related substrates (antiporters), but some display substrate–proton (symporters) transport activities or unidirectional (uniporters) [16,17,23,32]. It has been debated whether the mitochondrial carriers exist or function as a monomer or dimer. The crystal structures of ADP/ATP carriers exist in the IMM as monomers [26,102,103] and interact with respiratory supercomplexes in the presence of cardiolipin, a unique phospholipid found exclusively in the mitochondrion [104]. The formation of the homomeric and heteromeric complex may simply aid in assembly of the transport complex or act to avoid random, unfavorable protein–protein interactions in the packed environment of the inner mitochondrial membrane, such as the heteromeric complex of the mitochondrial pyruvate carrier [48] and dimeric structure of mitochondrial tricarboxylate carrier [50]. Many mitochondrial carrier proteins do not function as monomers, but have to assemble into oligomeric structures to become functional [52,53,82]. However, despite the fact that interactions between the mitochondrial carriers and mitochondria proteins are long known, they are not well studied. 

Nucleotide transporters include the mitochondrial ADP/ATP carrier, ANT, which we discussed above. Many claims have been made for the association of the ANT carrier with other proteins or protein complexes [34]. For a long time, it was thought that the ANT was a component of the permeability transition pore together with VDAC, BAX, and Bcl2 [25,105]. However, mitochondria lacking the carrier could still be induced to undergo permeability transition, resulting in release of cytochrome c [25,106], suggesting that this may not be the case. It has also been claimed that the ANT carrier, alongside a phosphate carrier, forms a complex with ATP synthase with a 1:1:1 stoichiometry—the so-called ATP synthasome [25,45]. However, the carriers are present in much higher numbers than ATP synthase to account for their relatively slow transport rates compared to the rate of ATP synthesis. Recent studies have revealed that ATP synthase forms rows of dimers on the ridge of cristae [25,107], a formation that is mediated by the Fo subunits of ATP synthase. This molecular arrangement leaves only the rotating c-ring exposed to the lipid bilayer, and thus there is no plausible binding site for carriers [25]. Furthermore, the ATP synthasome is supposedly stable in detergent, but affinity purifications of the carriers or ATP synthase under stabilizing conditions do not lead to detectable cross contaminations [25,108]. Therefore, it appears highly unlikely that the ATP synthasome exists. It has also been claimed that yeast ADP/ATP carriers (Aac2p) exists in physical association with the cytochrome c reductase (cytochrome bc1)–cytochrome c oxidase supercomplex and its associated TIM23 machinery, based on BN-PAGE and tagging experiments [25,109]. However, mitochondrial carriers are extremely unstable under BN-PAGE gel conditions in the absence of inhibitors, leading to aggregation and random association with other proteins [25,74]. 

In the course of the structural work of yeast ADP/ATP carrier, hundreds of milligrams of tagged Aac2p were purified under stabilizing conditions, but the copurification of other proteins or protein complexes was never observed [25]. Recently, it was claimed that the ANT carrier is associated with the translocase of the inner membrane (TIM) in a 1:1 stoichiometry [25,110]. These claims were based on SILAC experiments, but under the same conditions other mitochondrial carriers are also associated with TIM, but at lower stoichiometric ratios. The latter result indicates that the association is nonspecific, reflecting differences in expression levels or stability between carriers. The mitochondrial ANT carrier is one of the most abundant membrane proteins in the mitochondrion. In addition, the protein is very hydrophobic, leading to its structural instability in detergents. Moreover, as for the Aac2p described above, the protein does not protrude much from the membrane, and the surfaces that are exposed would appear to provide no opportunity for stable interactions [25]. Furthermore, there are no conserved residues on the surface of the carrier or asymmetrical features that are compatible with specific interactions with other proteins [25,37,111] Finally there are no plausible functional roles for these interactions, and the purported protein interaction partners are most likely experimental artefacts [25]. 

Inorganic ion transporters include the mitochondrial phosphate carrier PIC (SLC25A3) and the uncoupling protein UCP1 (SLC25A7). Whilst strong evidence supports divergent roles for UCP, both in dissipating proton gradients [112,113] and transporting aspartate, glutamate, and cysteinsulfinate [114], PIC has been characterized to transport phosphate in symport with protons for the synthesis of ATP [115,116]. The PIC was suggested in protein complex analysis to interact with mitochondrial peptidyl-prolyl cis-trans isomerase [46]; however, the work has not yet been followed up. Similarly in plants, a novel mitochondrial phosphate carrier interacts with NaStEP (a proteinase inhibitor essential to self-incompatibility) and mediates in the self-incompatibility of the pollen [47], whilst the TCA cycle enzymes citrate synthase, isocitrate dehydrogenase, and oxoglutarate dehydrogenase interact with a putative phosphate transporter in the AP-MS [83] as well as other TCA cycle enzymes to associate as the metabolon. Whilst follow up studies of these interactions were also not carried out, such interactions are not without precedence. Indeed, in the early work of Srere and coworkers looking into enzyme and enzyme protein assemblies, several such interactions were noted, including malate dehydrogenase, citrate synthase, and aconitase, on the basis of structure modelling studies [117,118]. Furthermore, although not a member of the MCF, the mitochondrial pyruvate carriers transport pyruvate to fuel the tricarboxylic acid cycle and thereby support ATP generation. However, this import also serves as a link to anabolic pathways for lipid and amino acid biosynthesis and gluconeogenesis. Heterocomplexes of MPC have been monitored by BRET [48] and purified following size exclusion chromatography [49]. The heterodimer is the active mitochondrial pyruvate carrier, while individual MPC proteins assemble as nonfunctional homodimers (Table 1) [49]. 

The situation is similar for other carriers of the MCF. For example, the tricarboxylate carrier catalyzes an electroneutral exchange of the dibasic form of a tricarboxylic acid (citrate, isocitrate, and cis-aconitate) with a proton for another tricarboxylate-H+, dicarboxylate (malate and succinate), or phosphoenolpyruvate [50,119,120,121]. Interestingly, the dimeric form of the tricarboxylate carrier protein was found when eel liver mitochondria were solubilized with the mild detergent digitonin [50]. However, the physiological relevance of this is, as yet, has not been unequivocably resolved. Similarly, the ubiquitously expressed dicarboxylate carrier protein (DIC) was found to exist in the dimeric form after lysis with the mild detergent digitonin using Western blotting, whereas only the monomeric form of DIC was discovered after lysis with SDS [51]. That said, the dimeric structure of the mitochondrial oxoglutarate carrier (OGC) has been demonstrated by the cross-linking and BN-PAGE of digitonin-lysed mitochondria [52,53]. The same is likely true for the dicarboxylate, tricarboxylate, and oxoglutarate carrier proteins, since these will certainly follow the same mechanism of the ADP/ATP carrier that was recently convincingly proven to be a functional monomer [38]. A note of caution in interpreting results from such studies can be provided by the case of UCP. For many years, it was thought that UCP1 formed a homodimer based on analytical ultracentrifugation [122], nucleotide binding [123], and protein cross-linking studies [124]; however, on the basis of size exclusion chromatography [125], UCP is now believed to act as a monomer that binds one purine nucleotide molecule [126]. Arguably more exciting, it has recently been reported that the human glutamate transporter, EAAT5, contains a C-terminal consensus motif that could interact with synaptic proteins that promote ion channel clustering [54]. Moreover, glutamate can also transport into mitochondria through the aspartate/glutamate carrier (AGC), combining the input of glutamate with the release of aspartate [36,39,55]. Size-exclusion chromatography has shown that two molecules of the aspartate/glutamate carrier are linked together by their N-terminal regulatory domains as the dimer (Table 1) [41]. Finally, the transporter activity of the MCFs proteins can perhaps also be modulated by forming complexes with interacting partners, such that the activity of some MPCs may be dependent on the formation of heterocomplexes. An example of this is the fact that an antiapoptotic protein, Bcl-2, was found to be an interacting protein partner of rat OGC, and when coexpressed with OGC in CHO cells, the total mitochondrial glutathione content was significantly increased 24 h post-transfection [127].

## 5. Conclusions and Perspective 

Despite the fact that complex formation of MCFs may play an essential role in metabolite import, MCF protein interactions have been poorly studied in living cells—most likely due to methodological limitations. Although both BN-PAGE and size exclusion chromatography have been used to analyze the heterodimers and homodimer of the MCFs, these dimers were suggested to be functional monomers, especially the ADP/ADP carriers [44]. Moreover, improved BRET has been used to monitor the MPC activity in real time, thus giving the possibility to use it for other MCFs [48]. BRET has been successfully used to monitor the heteromeric complex of the mitochondrial pyruvate carrier [48,49]. Interestingly, the activity of MPC in cancer cells could be monitored in the presence of various metabolites, including lactate, malate, and citrate. This method could be wildly used to monitor the assembly of mitochondrial carrier protein as several metabolite transporters may regulate their activity by the complex assembly. However, currently, these methods have not been applied to detecting protein–protein interactions of mitochondrial transport proteins such as MPC and MCFs. Given that physiological studies of the mitochondrial transport proteins are limited, the identification of interactors will help us to understand the mechanism of the mitochondrial transport proteins in vivo.

Moreover, the identification of PPI binding sites will be an important step here. One should note that some, albeit limited, physical evidence exists of MCF protein interactions, but little information is available concerning their biological function, and most examples of this are restricted to signal transduction. Indeed, given the limited exposure of the proteins due to the transmembrane domains, this may prove easier than for the metabolons that are considerably better characterized, such as those of glycolysis [6], the TCA cycle [1,84], and plant specialized metabolism [88]. In addition, the yeast enolase-associated macromolecular complex was identified to contain several mitochondrial carriers, including a phosphate carrier and an ADP/ATP carrier (Figure 3; [128]). As this example illustrates, the possibility of cross subcompartment metabolons exists, potentially suggesting a novel route of metabolic coordination. Given that glycolytic enzymes associated at the OMM of mitochondria form a metabolon [4,62,88,129], the mitochondrial transporters, such as VDAC, may also join the metabolon to improve the metabolite transport efficiency (Figure 3). Since many recent advances have been made in the study of metabolons [88,117], we believe the time is ripe to apply these to study the intriguing prospect that members of the MCF are involved in such structures and other types of protein–protein assemblies 

## Figures and Tables

**Figure 1 biomolecules-10-01107-f001:**
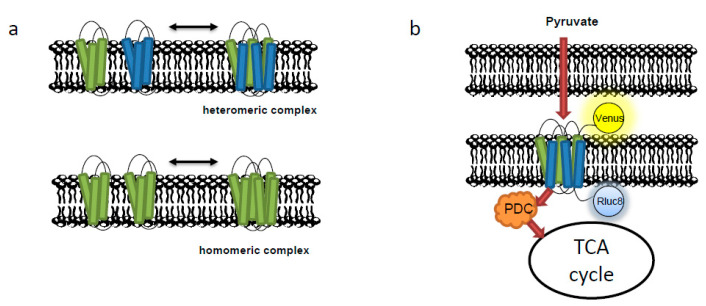
Protein complex of the mitochondrial carriers. (**a**) The association of both homodimers and heterodimers of the mitochondria carrier may result in the active transporter. (**b**) Monitoring the heterodimer of the pyruvate carrier by the bioluminescence resonance energy transfer (BRET).

**Figure 2 biomolecules-10-01107-f002:**
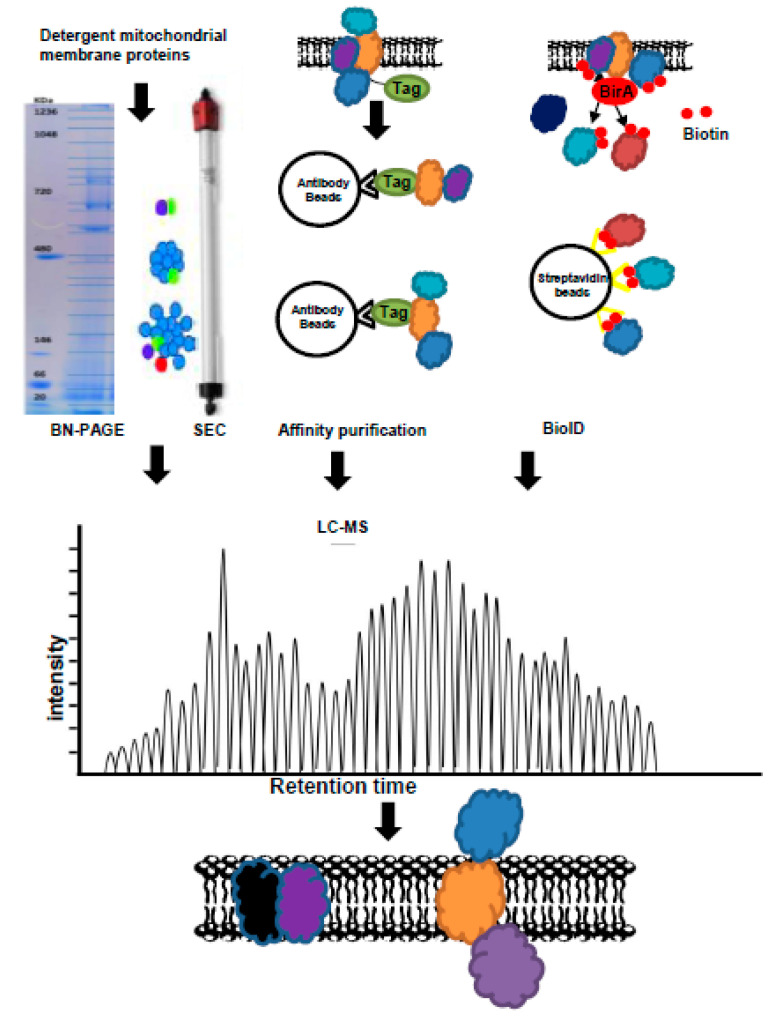
Methods used for protein–protein interaction combined with liquid chromatography–mass spectrometry (LC-MS). BN-PAGE: Blue native polyacrylaminde gel electrophoresis, SEC: Size exclusion chromatography, BioID: proximity-dependent biotin identification.

**Figure 3 biomolecules-10-01107-f003:**
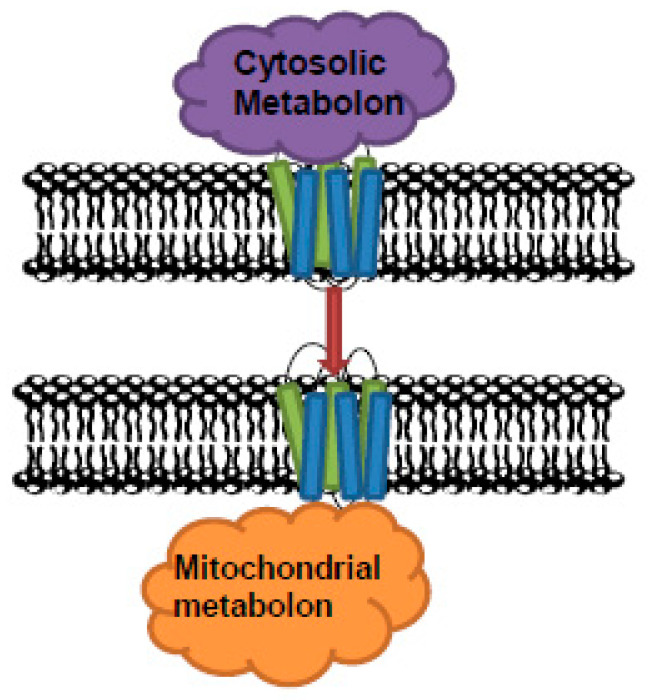
Metabolon associated to the mitochondrial carriers. The mitochondrial transport proteins may also join in the metabolon to improve the metabolite transport efficiency.

**Table 1 biomolecules-10-01107-t001:** The interactome of mitochondrial carrier family members and other mitochondrial transporter proteins.

Mitochondria Carrier	Interactor	Reference
VDAC	Tubulin, Desmin, Vimentin, plectin, hexokinase and creatine kinases, MtCK, ANT, cardiolipin	[10,42,43]
Acyl-dihydroxyacetone phosphate reductase	Unclear	[13]
ADP/ATP carrier	Previous suggested as dimer, and now convincedly proved as monomers	[44]
Phosphate carrier	ATP synthase, mitochondrial peptidyl-prolyl cis-trans isomerase, NaStEP, Citrate synthase, isocitrate dehydrogenase and oxoglutarate dehydrogenase	[25,45,46,47]
Pyruvate carrier	Heterodimer	[48,49]
Tricarboxylate carrier	Homodimer	[50]
Dicarboxylate carrier	Homodimer	[51]
Oxoglutarate carrier	Homodimer, BCL2	[52,53]
Glutamate transporter	Synaptic protein	[54]
Aspartate/glutamate carrier	Homodimer	[36,39,55]

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
