# Peer review of "On the Detection and Functional Significance of the Protein–Protein Interactions of Mitochondrial Transport Proteins"

_biomolecules, 2020, doi:10.3390/biom10081107_

Round 1
Reviewer 1 Report
In this review article, the authors detailed the available methods for the detection of protein-protein interactions of mitochondrial carriers as well as the functional significance of these interactions. Authors comprehensively cover the BRET, BiFC, BN-PAGE, chromatography, and proximity labeling based methods for the identification of protein-protein interactions and detailed their specific use in terms of mitochondrial carrier biology. Authors further detailed the interactome of the mitochondrial membrane carrier proteins.
Specific Comments:
- Most of the discussed topics in the review describe what is known, but critical inputs are not provided. The author may discuss the known fact for each topic, then they should provide their own perspectives on each topic, how future research may further advance this area of research. Authors should highlight the unanswered questions in the mitochondrial carrier proteins.
- It will be useful to add a table for the carrier proteins interactome.
- Numerous spelling mistakes throughout the manuscript. For e.g., Line 20. “Carried family”, it should be carrier family. The manuscript needs a thorough revision in terms of typos and grammar.
Author Response
- Most of the discussed topics in the review describe what is known, but critical inputs are not provided. The author may discuss the known fact for each topic, then they should provide their own perspectives on each topic, how future research may further advance this area of research. Authors should highlight the unanswered questions in the mitochondrial carrier proteins.
Response: We added a perspective part in the text to improve this part as suggested
- It will be useful to add a table for the carrier proteins interactome.
Response: We added this as Table I in the text
- Numerous spelling mistakes throughout the manuscript. For e.g., Line 20. “Carried family”, it should be carrier family. The manuscript needs a thorough revision in terms of typos and grammar.
Response: We changed these throughout the text.
Reviewer 2 Report
The manuscript of Zhang and Fernie is one of the most interesting manuscripts I have read recently. The discussed issue that the authors selected is very contemporary, detailed, and useful. However, the manuscript is hard to follow mostly because of dense text. The manuscript would benefit from revision of the scientific writing, especially revision of the paragraphs (shortening, dividing). Rules for paragraph coherence could be found online or discussed with the language department, if available. I am sure that if the text is improved from the stylistic point of view the scientific information will be more understandable and the review will have great citation potential.
minor comments:
correct:
page1 line 28 "such as"
page 2 line 38 research is always plural
page 4 line 129 "(citation)"
page 7 line 255 "ref needed"
Author Response
The manuscript of Zhang and Fernie is one of the most interesting manuscripts I have read recently. The discussed issue that the authors selected is very contemporary, detailed, and useful. However, the manuscript is hard to follow mostly because of dense text. The manuscript would benefit from revision of the scientific writing, especially revision of the paragraphs (shortening, dividing). Rules for paragraph coherence could be found online or discussed with the language department, if available. I am sure that if the text is improved from the stylistic point of view the scientific information will be more understandable and the review will have great citation potential.
Response: Thank you for the suggestion we have improved this. I would like to note however that the senior author is a native speaker with over 700 published papers so should not have to refer to the web for such information.
minor comments:
correct:
page1 line 28 "such as"
changed
page 2 line 38 research is always plural
changed
page 4 line 129 "(citation)"
changed
page 7 line 255 "ref needed"
changed
Round 2
Reviewer 2 Report
The style has been improved but I still find the paragraphs very long and there are some minor changes that should be still addressed.
please define abbrev for MCF, only MC is defined, not MCF
Line 100 - add space in front of MCF
chapter 4 - check the line spacing
Author Response
The style has been improved but I still find the paragraphs very long and there are some minor changes that should be still addressed.
Response: Thanks for the suggestion. Line 149, Line 248, line282, Line319, line404, line 440, line478
please define abbrev for MCF, only MC is defined, not MCF
Changed line 58
Line 100 - add space in front of MCF
changed
chapter 4 - check the line spacing
changed